# Future Approaches for Treating Chronic Myeloid Leukemia: CRISPR Therapy

**DOI:** 10.3390/biology10020118

**Published:** 2021-02-04

**Authors:** Elena Vuelta, Ignacio García-Tuñón, Patricia Hernández-Carabias, Lucía Méndez, Manuel Sánchez-Martín

**Affiliations:** 1Departamento de Medicina, Universidad de Salamanca, 37007 Salamanca, Spain; elena.vuelta.r@gmail.com (E.V.); patri_galle@usal.es (P.H.-C.); mendez_lucia@usal.es (L.M.); 2Institute de Investigación Biomédica de Salamanca (IBSAL), 37007 Salamanca, Spain; ignacio.tunon@usal.es

**Keywords:** CML, CRISPR/Cas9, *BCR/ABL1*, genome editing

## Abstract

**Simple Summary:**

In the last two decades, the therapeutic landscape of several tumors have changed profoundly with the introduction of drugs against proteins encoded by oncogenes. Oncogenes play an essential role in human cancer and when their encoded proteins are inhibited by specific drugs, the tumoral process can be reverted or stopped. An example of this is the case of the chronic myeloid leukemia, in which all the pathological features can be attributed by a single oncogene. Most patients with this disease now have a normal life expectancy thanks to a rationality designed inhibitor. However, the drug only blocks the protein, the oncogene continues unaffected and treatment discontinuation is only an option for a small subset of patients. With the advent of genome-editing nucleases and, especially, the CRISPR/Cas9 system, the possibilities to destroy oncogenes now is feasible. A novel therapeutic tool has been developed with unimaginable limits in cancer treatment. Recent studies support that CRISPR/Cas9 system could be a definitive therapeutic option in chronic myeloid leukemia. This work reviews the biology of chronic myeloid leukemia, the emergence of the CRISPR system, and its ability as a specific tool for this disease.

**Abstract:**

The constitutively active tyrosine-kinase *BCR/ABL1* oncogene plays a key role in human chronic myeloid leukemia development and disease maintenance, and determines most of the features of this leukemia. For this reason, tyrosine-kinase inhibitors are the first-line treatment, offering most patients a life expectancy like that of an equivalent healthy person. However, since the oncogene stays intact, lifelong oral medication is essential, even though this triggers adverse effects in many patients. Furthermore, leukemic stem cells remain quiescent and resistance is observed in approximately 25% of patients. Thus, new therapeutic alternatives are still needed. In this scenario, the interruption/deletion of the oncogenic sequence might be an effective therapeutic option. The emergence of CRISPR (clustered regularly interspaced short palindromic repeats) technology can offer a definitive treatment based on its capacity to induce a specific DNA double strand break. Besides, it has the advantage of providing complete and permanent oncogene knockout, while tyrosine kinase inhibitors (TKIs) only ensure that *BCR-ABL1* oncoprotein is inactivated during treatment. CRISPR/Cas9 cuts DNA in a sequence-specific manner making it possible to turn oncogenes off in a way that was not previously feasible in humans. This review describes chronic myeloid leukemia (CML) disease and the main advances in the genome-editing field by which it may be treated in the future.

## 1. Clinical Features of Chronic Myeloid Leukemia

Chronic myeloid leukemia (CML) is a myeloproliferative disease with an incidence of 1–2 cases per 100,000 each year, accounting for 15% of all new cases of leukemia [1]. The frequency is higher among adults, in whom the mean age of incidence is about 55 years, and indeed, rarely arises during childhood. It may affect both sexes, but is slightly more common in males, with a ratio of 2.2 men to 1.4 women per 100,000 affected [1,2]. The most common clinical symptoms of CML include fatigue, anemia, splenomegaly, abdominal pain, and recurrent infections. However, a large proportion of asymptomatic patients are diagnosed after an unrelated medical examination [1]. Three clinical phases of its pathological evolution are recognized. At first, CML disease is characterized by a myeloid hyperplasia in an indolent chronic phase (CP). At this point, leukemic stem cells (LSCs) respond to growth factors, but myeloproliferative differentiation pathways acquire an advantage because they are the main cause of the massive myeloid expansion characteristic of CML [3]. In this initial phase, myeloid progenitors and mature cells accumulate in the blood and extramedullary tissues. Without effective therapy, CML progresses through a period of increasing instability known as the acceleration phase (AP), ending in an acute leukemic-like disease known as the blast crisis phase (BP). The definitions of AP and BP are largely dependent on the proportion of blasts in the blood and bone marrow. AP and BP are characterized by a maturation arrest in the myeloid or lymphoid lineage, and newly accumulated genetic and epigenetic aberrations occur in LSCs [4]. The final BP stage can result in a lymphoblastic (25%), myeloblastic (50%), or biphenotypic/undifferentiated acute leukemic phenotype (25%), which indicates a stem origin for CML disease [5], as shown in Figure 1. Finally, bone marrow failure due to a lack of cell differentiation, and a massive infiltration by immature blasts, causes patient mortality from infection, thrombosis, or anemia [6].

Diagnosis is based on detecting the hallmark of CML, the presence of the chromosome 22 abnormality known as Philadelphia (Ph), named after the US city in which it was first observed. It is the result of the reciprocal translocation between chromosomes 9 and 22-t(9;22) [7]. Conventional cytogenetics, fluorescence in situ hybridization (FISH), and reverse transcription PCR (RT-PCR) are the techniques commonly used to confirm a diagnosis of CML and to evaluate the response to therapy.

Before successful treatments became available, the median survival of CML patients after diagnosis was approximately 3–5 years [8,9]. The therapeutic landscape of CML changed profoundly with the introduction of tyrosine kinase inhibitor (TKI) drugs [8,10,11], and most patients with CP-CML now have a normal life expectancy. However, treatment discontinuation is only an option for a small subset of patients [12].

## 2. Molecular Biology of Chronic Myeloid Leukemia

Nowell and Hungerf, in 1960, first described the Ph chromosome, a small chromosome present in the bone marrow cells of CML patients [7]. It was the first time that a chromosomal abnormality had been linked to a particular neoplasia [13]. Subsequent investigations confirmed that the generation of the Ph chromosome was due to the t(9;22) (q34;q11) translocation. The next breakthrough in our understanding of CML occurred in the 1980s, when it was demonstrated that this rearrangement gave rise to a fusion gene [14]. In this translocation, the analogue of the v-*ABL* protooncogene from chromosome 9 is moved to the breakpoint cluster region of the *BCR* gene on ch22. The location of the breakpoints between the two loci is variable [15]. Commonly, the breakpoint at the *ABL* locus occurs in a DNA region spanning more than 200 kb housing exon 2. At the *BCR* locus, the breakpoints occur in the major breakpoint cluster region (M-*bcr*), which spans a 3 kb region that includes exons 13 and 14 of *BCR*. All the rearrangements involving both breakpoint regions give rise to a 210 kDa protein, the most common chimeric transcript in CML [16]. However, in a minority of CML cases, the *BCR* breakpoint is located near exon 2, termed the minor breakpoint cluster region (m-*bcr*). In these cases, the resulting mRNA gives rise to a 190 kDa protein [15]. Finally, another infrequent breakpoint cluster region (μ-*bcr*) exists, downstream of *BCR* exon 19, which generates a 230 kDa protein when it is translocated to the *ABL1* locus [17], as shown in Figure 2.

Since the *BCR/ABL1* fusion was described, the efforts of the scientific community have focused on elucidating its molecular roles in CML pathology. Several studies have shown the aberrant and constitutive tyrosine kinase activity of the *BCR/ABL1* oncoprotein, highlighting this activity as being responsible for the transformation of the hematopoietic stem cell [18,19,20,21]. The fusion of the two genes constitutively activates the tyrosine kinase domain of *ABL1*, which contains three SRC homology domains (SH1–SH3). The SH1 domain enables the tyrosine kinase function, whereas the SH2 and SH3 domains mediate interactions with other proteins [22]. The SH3 domain is critical to the regulation of *ABL1* kinase activity, thereby presenting a target for clinical therapy. It is known that the fusion between the 5′ end of *BCR* and the SH3 domain of *ABL1* abrogates the physiological suppression of the kinase [23]. Meanwhile, *BCR* has an important coiled-coil (CC) domain that will allow *BCR/ABL1* dimerization and subsequent trans-autophosphorylation, thus increasing the molecular signal [24], as shown in Figure 3. The phosphorylation of the Y-177 tyrosine residue domain SH2 of *ABL* allows the high-affinity binding of the growth factor receptor-bound protein 2 (GRB2) as well as the scaffolding protein Gab2, activating the Ras pathway [25]. This aberrant kinase signaling activates many target proteins, such as the PI3K, AKT, JNK, and SRC family kinases, as well as transcription factors such as STATs, nuclear factor-κB, and MYC [26,27,28].

The constitutively active signaling causes cell reprogramming and expansion of the LSC clone. As a result, *BCR/ABL1*-positive hematopoietic stem cells exhibit uncontrolled proliferation [29], lack of response to apoptotic signals [30], alterations in cell adhesion [31], impaired differentiation [32], and independence of growth factors [33]. As a consequence, a myeloid differentiation bias is commonly observed in the chronic phase of CML.

## 3. Conventional Therapies for Chronic Myeloid Leukemia

The history of CML treatment can be considered one of the great milestones of modern cancer medicine. From its discovery until the 1980s, the standard treatment for CML consisted of conventional chemotherapy. Arsenic was the first treatment to be administered in the 19th century, but was superseded by alkylating drugs such as busulfan and hydroxyurea in the 1960s [34,35]. Unfortunately, they did not delay the onset of disease progression and facilitated only a modest improvement in survival. The introduction of interferon-α in the 1970s induced complete cytogenetic remission in 10–15% of patients, and increased median survival to 6 years [36]. However, interferon-α treatment has serious side-effects, and treatment had to be discontinued in most patients, causing them to relapse. In this context, allogenic stem cell transplantation (SCT) was the only therapeutic option that could provide increased long-term survival, and so it became the first-line treatment in the 1990s for patients in the chronic phase [37,38,39]. Even today, this therapeutic option is the only one with the potential to definitively cure CML patients in this phase. The SCT procedure involves bone marrow ablation (by chemotherapy or radiotherapy) followed by the infusion of normal allogenic stem cells. However, it is only available to a small number of patients who have an HLA-matched donor, and is associated with a significant transplant-related mortality rate [39]. Nowadays, SCT is used solely as a last-resort salvage option.

As mentioned above, CML is a type of cancer in which all the pathological features can be attributed to a single genetic event, in this case the *BCR/ABL1* fusion. Knowing that the tyrosine kinase activity of *BCR/ABL1* is essential for the malignant transformation of cells, the search for compounds that inhibit this activity became imperative. During the 1990s, various tyrosine kinase inhibitors (TKIs) were tested to evaluate their therapeutic potential in CML [40,41]. The mechanism of action of these compounds is based on competition with adenosine triphosphate (ATP) or the protein substrate of the kinase, whereby *BCR/ABL1* activity is inhibited at the protein level. Finally, in the 2000s, the Novartis compound STI571 (later known as imatinib mesylate), which showed surprising results by selectively inhibiting *BCR/ABL1* at micromolar concentrations, was approved as therapy for CML [42,43]. The arrival of TKIs marked a watershed in the treatment of CML and they remain the frontline therapy for CML. Thanks to TKIs, CP-CML patients, who, before 2001, had a survival rate of 20% at 8 years, now have a rate of 87%, and a life expectancy like those of healthy people of the same age [10,11,17]. Despite the success achieved with TKI-based treatments, there are still obstacles to overcome. The main concern is that TKI drugs do not tackle the etiological cause of CML and the oncogenic event remains uncorrected/unedited. The existence of residual *BCR*/*ABL*-positive cells, which remain “oncogenic-quiescent”, has been demonstrated, indicating that TKIs do not completely eliminate the LSCs [12]. TKIs efficiently silence the oncogenic activity of *BCR*/*ABL* while the drug is present, but the remaining LSCs can lead to relapse after TKI therapy ceases, as shown in Figure 4.

In this scenario, lifelong oral medication is necessary, and treatment discontinuation is only an option in those patients who were able to achieve and maintain strong molecular responses. Lifelong administration facilitates adverse effects in many patients and a significant percentage of them eventually become resistant to TKI treatment [44]. The identification of various forms of resistance has led to the development of second- and third-generation TKIs that are effective against kinase-specific mutations in these patients [45]. Taking this therapeutic scenario into account, it is still necessary to seek new and definitive alternative therapies. Currently, any coding sequence can be abolished by CRISPR (clustered regularly interspaced short palindromic repeats)/Cas9 nucleases [46,47,48] or zinc finger nuclease [49], which means there is an opportunity of a definitive cure available to TKI-resistant CML patients. Thus, CRISPR/Cas9 system could be a definitive therapeutic option.

## 4. Genome-Editing Nucleases for Gene Therapy

Advances in molecular biology and genetics in recent years have broadened our knowledge of genetically based diseases, and very many genes involved in their development have been identified. These same advances have made it possible to develop the genome-editing technology with which these candidate genes can be genetically manipulated. With the advent of engineered chimeric proteins with nuclease activity, such as zinc-finger nucleases (ZFNs) and transcription activator-like effector nucleases (TALENs), genome manipulation has become more feasible than ever [50,51]. These new approaches overcome the difficulties associated with previous genome-editing techniques based on homologous recombination (HR), such as low efficiency, and laborious and time-consuming assays [52]. The mechanism of action of genome-editing nucleases is based on the generation of double-strand breaks (DSBs) in the DNA that stimulate the endogenous cellular DNA repair mechanisms: non-homologous end-joining (NHEJ) and homology-directed repair (HDR). NHEJ results in the introduction of random insertion or deletion (indel) mutations that, in a coding sequence, most frequently lead to frameshift mutations that generate null alleles. The HDR pathway exploits the phenomenon of homologous recombination specifically to introduce an exogenous donor DNA template in the DSB site, allowing mutated sequences to be replaced or edited [53], as shown in Figure 5.

ZFN and TALEN have been widely used for decades, but the proteinaceous nature of their structure leads to serious technical drawbacks, such as the complexity of design and high costs [54]. Fortunately, the recent development of the CRISPR/Cas9 system in the genome-editing field has revolutionized this methodology. The simplicity of this system offers a powerful, effective, low-cost, and universal tool heralding a new era for gene therapy.

## 5. Overview of the CRISPR/Cas9 System

Most of archaea and almost 50% of bacteria have an adaptive immune system to defend them against phage infection. This system is defined by a genomic locus with a series of short palindromic repeats separated by unique “spacers”, preceded by an AT-rich “leader” sequence and forming a cluster [55]. Francisco Mojica was the first researcher in 1993 to describe this matrix of tandem-repeated sequences working on the archaea *Haloferax mediterranei* [56]. Previously, a similar structure was described in *Escherichia coli* and he also spotted a connection with eubacteria [57]. Mojica coined the acronym of CRISPR (clustered regularly interspaced short palindromic repeats) in accordance with Ruud Jansen, who first used the term in print in 2002 [58]. Three years later, it was reported that these unknown spacer sequences had a high percentage of similarity with sequences found in various types of bacteriophages and plasmids and it may represent a immunological memory [59,60,61].

In 2007, Barrangou et al. demonstrated that the CRISPR system was a rudimentary prokaryotic immune system that protects prokaryotes against foreign DNA infections [62]. They also studied the role of the CRISPR-associated proteins (Cas) Cas7 and Cas9, suggesting that Cas7 was involved in generating new spacers and repeats and Cas9 in breaking the DNA [63]. Later it was showed that the different spacers are interspersed with tandem sequences and they are expressed as small guide CRISPR RNAs (crRNAs). These crRNAs were responsible for CRISPR-based resistance and they can be transferred from a resistant to a naïve strain, inducing resistance in the second. crRNAs drive the Cas9 protein to cleave the invader genome, and this find opened the door to direct the destruction of a DNA sequence like a restriction enzyme but in a specifically addressable manner [64,65,66,67]. Importantly, the only requirement for Cas9 nuclease activity was the existence of a small PAM motif (protospacer adjacent motif) at the 3′ end of the target sequence [68,69,70]. A single precise blunt-end cleavage event three nucleotides upstream of the PAM sequence was the consequence of the Cas9 nuclease activity.

The transactivating CRISPR RNA (tracrRNA) completed the puzzle to clarify the nature of Cas9 activity [71]. The tracrRNA is a scaffold that partially hybridizes with the crRNA and the Cas9 endonuclease, allowing all the components to be assembled [48], as shown in Figure 6.

To further simplify the system, Charpentier and Doudna showed that the two RNAs could function in vitro when fused into a single-guide RNA (sgRNA). Like Siksnys [72], they showed that Cas9 could cut purified DNA in vitro and it could be programmed with custom-designed sgRNAs [48].

These discoveries and the demonstration of their in vitro activity opened the door to using this system as a genome-editing nuclease [73]. However, the question that remained to be addressed was whether the CRISPR system would also work in mammalian cells. By mid-2012, Feng Zhang designed a robust three-component system consisting of Cas9 (orthologs from *Streptococcus thermophilus* or *S. pyogenes*), tracrRNA, and a CRISPR array for targeting 16 sites in the human and mouse genomes. Zhang answered the question and he showed that it was possible to mutate one or several genes at the same time, inducing indels by NHEJ. Besides, the CRSPR system could insert new sequences at the target site via HR when a repair template was added [74,75]. Its simplicity, effectiveness, and universality mean that the CRISPR/Cas9 system has rapidly become the preferred tool for RNA-guided genome editing. In fact, it has been widely applied for gene modification in several model systems [76,77,78,79,80]. In 2006, the emergence of the “base editing” era, a new approach that enables the direct, irreversible conversion of one target DNA base into another in a programmable manner, without requiring DSBs or donor templates, was developed from David Liu’s lab. An engineered fusion of CRISPR/Cas9 with the cytidine deaminase enzyme has the ability to mediate the direct conversion of cytidine to uridine, thereby effecting a C → T (or G → A) substitution [81]. The door to correct a variety of point mutations relevant to human disease began to open. However, it was reported that these DNA base editors can cause substantial off-target editing in both genomic DNA and RNA, and thus further studies will be performed to improve the selectivity of DNA base editors [82].

Despite system deficiencies that still need to be improved, it is likely that the CRISPR/Cas9 systems will be incorporated into the therapeutic strategy for the treatment of monogenically inherited disorders and malignancies whose pathological features can be attributed to a single genetic event, such as gene fusion [83,84].

## 6. New CRISPR-Cas Systems and Approaches

However, Zhang’s scientific contributions did not stop there and continued (and continue today) to revolutionize the field of gene editing. In 2015, Zhang’s team reported a novel and more accurate CRISPR system from *Acidaminococcus* and *Lachnospiraceae*, with efficient genome-editing activity in human cells, the Cas12a (Cpf1) [85]. The Cpf1 coding sequence is smaller than Cas9, it requires only one RNA (tracrRNA-independent), and generates sticky-end DSBs that are less prone to non-homologous end joining (NHEJ), being ideal for precise gene editing. Given these advantages, the Cpf1 system is more feasible to in vivo deliver and it could become a better gene-editing tool than Cas9 [86,87]. A peculiar property of Cas12a was that it completely degraded single strand DNA molecules after its activation. By combining Cas12a ssDNase activation with isothermal amplification, Doudna’s team created a method named DETECTR (DNA endonuclease-targeted CRISPR trans reporter), which achieves attomolar sensitivity for DNA detection, showing its ability to detect human papilloma virus in patient samples in a rapid and specific manner [88].

In 2017, Zhang and colleagues again astonished the scientific community, reporting a novel CRISPR system to target and edit RNA, the class 2 type VI CRISPR-Cas effector Cas13a [89,90]. The nuclease activity of this protein allows gene knockdown without genomic alteration. However, Cas13a cleaves all the RNAs around after it becomes enzymatically active following the first single strand RNA break. Far from being a disadvantage, this promiscuous feature was used for DNA or RNA detection in diagnostics [91]. Specific high-sensitivity enzymatic reporter unlocking (SHERLOCK) was the first platform based on CRISPR-Cas13 systems for rapid and specific detection of viruses or mutations in patient liquid biopsy samples [92].

## 7. CRISPR Gene Therapy in CML

In the last five years, the number of scientific papers reporting work on CRISPR/Cas9 in the context of leukemia research has increased enormously [84,93,94,95,96]. Many of them concern in vitro studies to clarify the role of a variety of genes in leukemia development [97]. These studies identify key genes that will subsequently be edited in leukemic cells using CRISPR/Cas9 technology.

In 2015, Valletta et al. demonstrated for the first time that the CRISPR/Cas9 system could correct acquired mutations in a human myeloid leukemia cell line [98]. CRISPR-Cas9 was then successfully used in animal models of genetic diseases. Finally, the first clinical trials involving CRISPR-Cas9 in humans were initiated in 2016 [99].

Focusing on hematopoietic stem cells (HSCs), the first clinical trial to treat thalassemia (NCT03655678) using CRISPR-Cas9-modified HSCs was approved in 2018 [100]. In this sense, CML could also be one of the best candidates with which to evaluate the therapeutic potential of the CRIPSR/Cas9 system. CML is an HSC malignancy directed by a single oncogene. The singularities of HSCs, which sustain the long-term generation of all hematopoietic lineages, make CML an ideal candidate for gene therapy. The special characteristics of self-renewing and multipotent HSCs imply that gene editing or ablation by CRISPR will be inherited by all daughter cells, restoring a new hematopoiesis. Furthermore, the peculiarities of the hematopoietic compartment, which make possible the collection and subsequent reinfusion of HSCs, enable the development of ex vivo therapies, and thereby the evaluation and selection of the edited HSCs, improving the safety and efficiency of the process.

Imatinib therapy is based on the knowledge that the *BCR/ABL1* fusion is the underlying cause of CML pathogenesis. For this reason, it is reasonable to surmise that the CRISPR/Cas9-induced gene interruption of *BCR/ABL1* might offer a definitive cure. Several studies have been recently performed to study the ability to disrupt the *BCR/ABL1* oncogene, showing the CRISPR-Cas9 system as a therapy tool ready to reach clinical trials in the near future, as shown in Table 1.

The development of immunodeficient mice for human HSC engrafting [101] and of mouse models that mimic human CML [102] has provided new opportunities to evaluate these CRISPR/Cas9 therapeutic applications. Recently, several in vitro and in vivo studies have explored the ability of CRISPR/Cas9 to destroy the *BCR/ABL1* gene fusion. In 2017, Garcia-Tuñón and coworkers demonstrated for the first time that the CRISPR/Cas9 system effectively abrogates the *BCR/ABL1* oncogene, reversing its tumorigenic activity [96]. They showed in a CML xenograft animal model how edited CRISPR cells lost their ability to proliferate and survive, and that no tumors developed when the edited cell was selected. Their results constituted the proof-of-principle that *BCR/ABL1* abrogation by the CRISPR system results in the loss of tumorigenicity.

In 2018, Wenli Feng’s group demonstrated that other genome-editing nucleases, like ZFN nucleases, achieved the abrogation of the *BCR/ABL1* oncogene [95]. Using a pair of ZFNs targeting the exon 1 of *BCR*, a premature stop codon was created triggering a truncated oncoprotein. The apoptotic rate was higher, and the proliferative capacity was lower in the ZFN-edited cells. The same group published a subsequent study in which they overcame the technical limitations linked to the use of the ZFNs [94]. The authors adopted a new strategy based on CRISPR RNA-guided FokI nucleases (RFNs) to target exon 2 of *ABL1*. According to them, the combination of the universality of the CRISPR site design and the specificity of the FokI cleavage would provide an efficient and secure editing tool that would avoid the limitations of previous systems, such as the labor-intensive design of ZFNs and off-targets of CRISPR/Cas9. RFN-editing proved to be effective, achieving a reduction in the expression of *BCR/ABL1* and its downstream targets, in the imatinib-sensitive and imatinib-resistant forms of K562. Edited cells showed a loss of their malignant potential, reflected in a depressed proliferative and colony-forming capacity in vitro. Furthermore, when these edited cells were transplanted by intravenous injection into the tail vein of immunodeficient NOD/SCID animals, they showed an impaired in vivo leukemogenic capacity.

Recently, new work focusing on the disruption of *BCR/ABL1* by genome-editing nucleases as a therapeutic strategy in CML has revealed the therapeutic potential of the CRISPR system. In 2020, Chia-Hwa Lee et al., using a CRISPR/Cas9 lentiviral vector to disrupt *ABL1* in the human CML K562 cell line, demonstrated a reduced proliferation rate as a consequence of *BCR/ABL1* disruption [103]. Ex vivo transduction of peripheral blood mononuclear cells from CML patients was performed to evaluate the therapeutic potential of this viral system in the clinical milieu. They observed a high rate of apoptosis in the transduced cells and demonstrated that the disruption of the *ABL1* non-rearranged allele did not trigger important consequences. The T-cell lineage was not affected by CRISPR activity at this *ABL1* non-translocated locus.

A new approach based on the use of two guides to induce a large deletion and selectively eliminate fusion oncogenes has been developed by Rodriguez-Perales and coworkers [84]. Combinations of two sgRNAs targeting *BCR* intron 8 and *ABL* intron 1 regions were designed to induce a 133.9 kb deletion on *BCR/ABL1* oncogene. This new strategy induced a frameshift alteration of the entire *ABL1* DNA-binding domain. For any given sgRNA combination, electroporated K562 cells showed a significant decrease (∼85%) in clonogenic capacity in vitro and an increase in apoptosis. Importantly, they used cord blood-derived human hematopoietic progenitor hCD34+ cells to study whether a side effect was produced because of the CRISPR-Cas9 activity. In vitro analysis of targeted CD34+ cells revealed no difference in proliferation in long-term culture, suggesting no production of collateral cancer-driven genomic alterations. Besides, K562 transduced cells with an adenoviral vector carrying all the CRISPR reagents were subcutaneously injected into immunodeficient mice resulting in an 88% decrease in tumor size compared with control tumors.

Finally, Vuelta et al. recently reported their design of a new CRISPR/Cas9 short-deletion system that efficiently interrupts the *BCR/ABL1* oncogene in murine and human cell lines and, for the first time, in primary leukemic stem cells Sca1+ from a CML mouse model and CD34+ from human CML patients [104]. They demonstrated that CRISPR/Cas9-edited LSCs had impaired tumorigenic activity and fully restored capacity for multipotency. Further, they showed that the infusion of CRISPR/Cas9-edited LSCs confer a significant therapeutic benefit on orthotopic patient-derived xenografts (PDXs) and on CML mouse models. They revealed that CRISPR/Cas9 technology can easily be used to destroy driver oncogenes like *BCR/ABL1*, providing proof-of-principle for gene therapy through genome-editing nucleases.

## 8. Future Directions

With the advent of genome-editing nucleases and, especially, the CRISPR/Cas9 system, the possibilities of modifying the genome of species have reached hitherto unimaginable limits. In this context, gene therapy is one of the fields that has experienced a great impulse. The possibility of definitively curing genetic diseases, by direct correction of the underlying cause of the pathology, has ceased to be a future possibility and become a current reality. However, certain limitations still hinder the use of gene therapy as part of routine medical practice. Like other gene therapy approaches, the greatest limitation of in vivo CRISPR therapy is the difficulty of finding an optimal and safe delivery method. On the other hand, the preexisting adaptive immunity to Cas9 proteins in humans [105] could be considered and new Cas proteins should be employed. The issue about CRISPR off-targets also needs to be resolved [106]. The CRISPR-Cas9 system induces DSBs at target sites in genomic DNA, but can also generate undesirable cleavages outside of on-target sites. Cleavage at off-target sites can trigger mutations which may result in the disruption of normal genes. Efforts to discover new Cas variants with high fidelity and a protospacer adjacent motif less restrictive than NGG sequence will offer soon a solution [107,108]. Finally, despite the development of new and increasingly efficient methods, 100% editing efficiency is unattainable. However, guaranteeing the absence of unedited cells is imperative in hematopoietic malignancies that are clinically treated, such as the disruption of *BCR/ABL1* in CML. A possible solution would involve the selection of the correctly edited cells, which would entail the design of genome-editing approaches that simultaneously allow the genetic correction and expression of a selectable cell marker.

In summary, the enormous therapeutic potential of the CRISPR/Cas tools have been widely corroborated in numerous research papers and in clinical trials. There are technical limitations associated with this technology, but the number of possible alternatives to overcome them has increased at the same rate. We are certain that CRISPR/Cas gene therapy will become a routine clinical practice in the near future.

## Figures and Tables

**Figure 1 biology-10-00118-f001:**
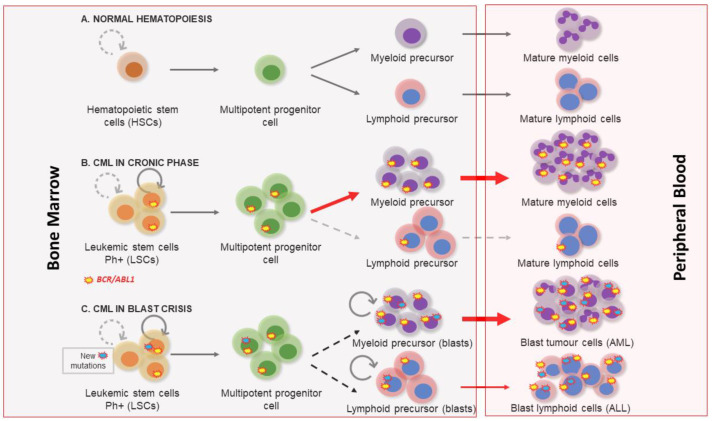
Chronic myeloid leukemia clinical phases. (**A**) Normal hematopoiesis characterized by the existence of hematopoietic stem cells with a controlled self-renewal and multipotency ability, resulting in balanced hematopoiesis between myeloid and lymphoid lineages. (**B**) In the chronic phase the myeloproliferative differentiation pathway acquires an advantage, and a massive myeloid expansion is produced. (**C**) Blast crisis is characterized by a maturation arrest in the myeloid or lymphoid lineage. Newly accumulated genetic and epigenetic aberrations appear in leukemic stem cells (LSCs) and blast cells go out from bone marrow to peripheral blood.

**Figure 2 biology-10-00118-f002:**
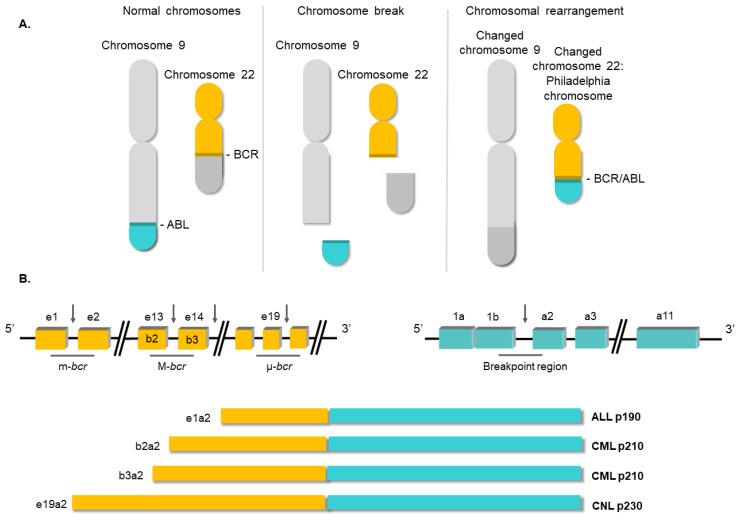
Structure of the *BCR/ABL1* oncogene. (**A**) Schematic representation of the t(9;22) (q34;q11) translocation triggering the Philadelphia chromosome. (**B**) Breakpoint locations between *BCR* and *ABL1* genes. Different fusion protein combinations yield different outcomes.

**Figure 3 biology-10-00118-f003:**
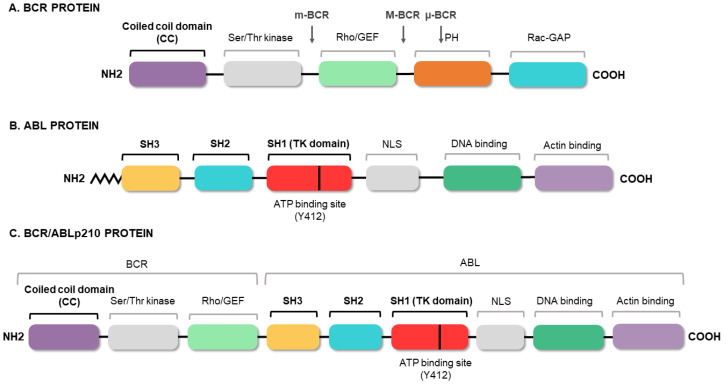
*BCR*/*ABL* protein domains. Protein regions located in the *BCR* (**A**) and *ABL* (**B**) proteins, and those maintained in the fusion (**C**). The figure highlights the coiled-coil (CC) domain of *BCR*, which allows the dimerization of the oncoprotein, and the three SRC domains of *ABL1*, including the tyrosine kinase domain (SH1) and the regulatory domains (SH2 and SH3).

**Figure 4 biology-10-00118-f004:**
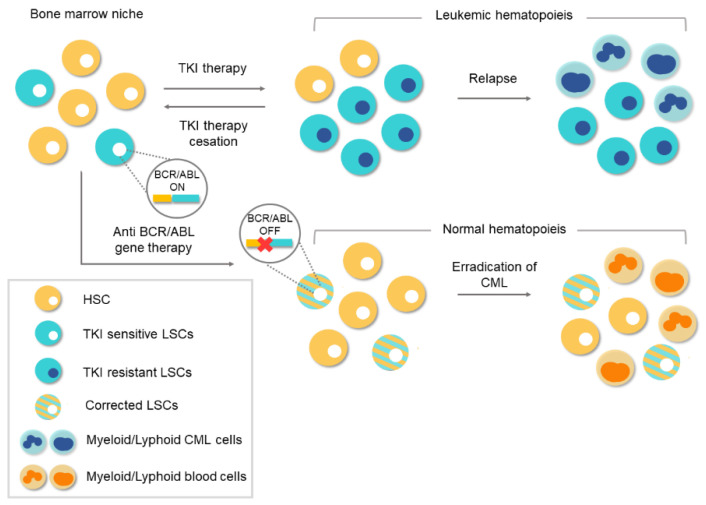
Conventional therapies vs. gene therapy for chronic myeloid leukemia (CML). Tyrosine-kinase inhibitor (TKI)-based conventional therapies are effective at silencing *BCR/ABL1* in leukemic stem cells (LSCs). Treatment cessation can lead to relapse because of the existence of residual *BCR/ABL1*-positive cells. The appearance of TKI-resistant LSCs during treatment can lead to a relapse of the disease. However, anti-*BCR/ABL1* gene therapy would eliminate the oncogene at the genome level. Corrected LSCs would be able to repopulate the bone marrow niche and thereby enable normal hematopoiesis.

**Figure 5 biology-10-00118-f005:**
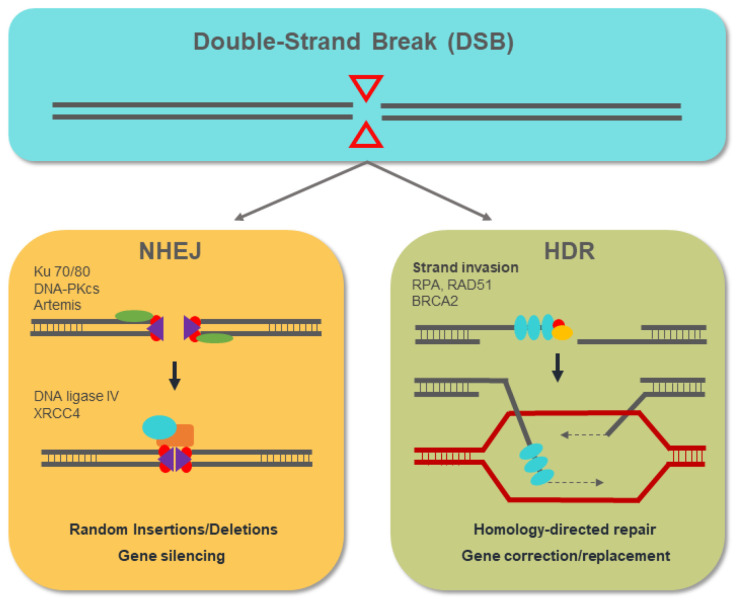
The non-homologous end-joining (NHEJ) mechanism involves the action of the proteins ku70/80, DNA-PKcs, and Artemis, with the ability to bind to the free DNA ends that are generated. The resected DNA ends are joined by the action of ligase IV with the insertion of a variable number of nucleotides (indels) that, in most cases, lead to the generation of null alleles. The homology-directed repair (HDR) pathway begins with the resection of the released DNA ends. The RPA, Rad51, and BRCA2 proteins act by binding and protecting the ssDNA that is generated. Through homologous recombination, the HDR pathway allows the introduction of DNA templates from exogenous donors at the double-strand break (DSB) site, replacing the target genomic sequence.

**Figure 6 biology-10-00118-f006:**
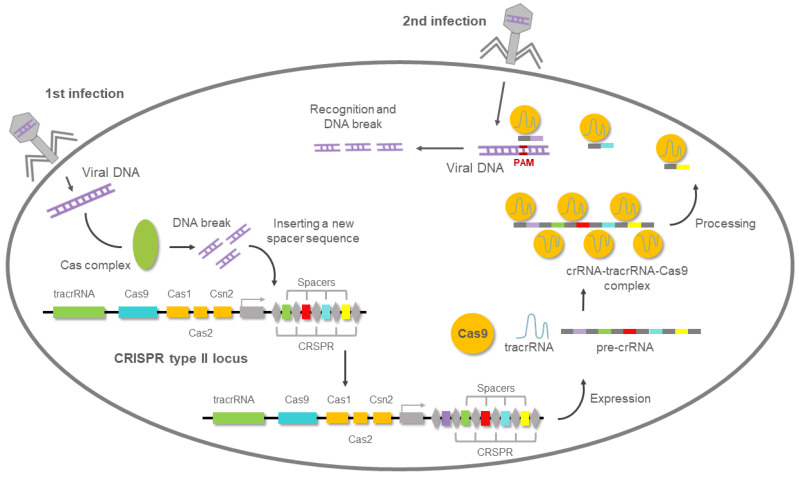
CRISPR (clustered regularly interspaced short palindromic repeats)-mediated adaptive immunity system presents in prokaryotes. After the first viral infection, the Cas complex excises the viral DNA, then introduces it into the bacterial genome. When the second viral infection occurs, a complementary RNA (crRNA) to that of the viral genome is used to guide the Cas9 nuclease to degrade the viral DNA.

**Table 1 biology-10-00118-t001:** Therapeutic strategies to disrupt the *BCR/ABL1* oncogene in CML by genome-editing nucleases.

Target	Cell Type	Genome Editing System	Outcomes	Reference
Fusion sequence	Boff p210 (mouse)	CRISPR/Cas9	Subcutaneous injection of edited single cell derived clones was unable to generate tumors in a CML xenograft model.	[96]
*BCR* exon 1	K562 (human) and patient derived CD34+ cells	ZFNs	Intravenous tail vein injection into NOD/SCID mice of the edited K562 showed a lower tumorigenic capacity in vivo. Lower proliferative capacity in vitro was observed in edited primary cells.	[95]
*ABL1* exon 2	K562 (human) and patient derived CD34+ cells	CRISPR RNA-guided FokI nucleases (RFNs)	Similar results to those of their previous work. High efficiency and greater security by reducing the frequency of off-targets, compared with CRISPR/Cas9 system.	[94]
*ABL1* exon 2	K562 (human) and peripheral blood mononuclear cells (PBMCs) of CML patients	CRISPR/Cas9	Virus-mediated *ABL1*-targeting to edit luciferase-labeled K562 into a systemic leukemia xenograft model. Bioluminescence imaging showed a significant reduction of leukemic cells in vivo.	[103]
Fusion sequence	K562 (human) and patient derived CD34+ cells	CRISPR/Cas9	Specific targeting of the *BCR/ABL1* fusion sequence with a pair of guides directed towards intronic sequences of each of the genes involved in the fusion that will cause a deletion in those cells that carry the translocation.	[84]
*ABL1* exon 6	Boffp210 (mouse), K562 (human), Lin- CML mouse model and patient-derived CD34+	CRISPR/Cas9	Edited HSCs from CML mouse model restored normal hematopoiesis in NOD/SCID bone marrow niche. Edited patient-derived CD34+ are capable of regenerating normal hematopoiesis in the bone marrow niche of NOD/SCID mice.	[104]

## Data Availability

Not applicable.

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
