# Peer review of "Future Approaches for Treating Chronic Myeloid Leukemia: CRISPR Therapy"

_biology, 2021, doi:10.3390/biology10020118_

Round 1
Reviewer 1 Report
General impression of the manuscript.
- The manuscript is well written.
- The manuscript adds to the work described by this group previously and it intends to give an overview of the use of Crispr therapy in the CML field.
- The balance is strongly towards CML description (approx. 6 pages) and less strongly on Crispr Cas technology (approx. last 2.5 pages).
- It would certainly contribute to the field when the information on Crispr Cas application in CML would be more extensive and the emphasis would be a bit less on the BCR ABL biochemistry and physiology.
Details concerning the main text and figures.
- In the abstract: change the last sentence to emphasize the focus on Crispr Cas technology and mention the reason why the technology is promising already here.
- Maybe addition of a few references, for instance Sternberg / Doudna on the off target effects of CAS9 would have added value for the complete story.
- Line 82, change enabling to ‘thereby presenting a target for clinical therapy’.
- Figure 3: text is missing in the manuscript on the upper side of the figure; the figure may not be well placed or the text is really absent.
- Line 127, the meaning of ‘destroyed’ is not clear in the context of the sentence (it is uncorrected or destroyed). Maybe remove the word destroyed.
- Figure 4. The information is clear. However, when the legend explains the repopulation potential of modified LSC’s, they are eradicated and thus absent in the last picture, the visual information is a bit misleading, since these LSC’s are now depicted as normal cells. I get the point, but maybe the figure can be more in line with the text in the legend.
- Figure 5. To my opinion, the green HDR panel is not so clear for the reader, maybe because of the arrows drawn on both strands. In addition, the figure has a few typos.
- Line 168 and others; the word finally is used quite often, whereas it is not always the end of the information presented. For instance, the Crispr Cas biotechnological revolution only starts after the mentioned hallmarks. I think in this respect that the paper by Wright, Nunez and Doudna in Cell should be referred to and there are other great papers that are left aside. This is the main point of my comments: the Crispr Cas part can be stronger and more informative.
- Line 185. This is an opportunity to refer to the review of Wu in theranostics on Crispr systems in human genetic disease.
- Line 207. The relevance of Table 1 is not explicitly mentioned, it is only referred to in the text. I think it would be important to explain the information presented in the table.
- Line 242. Please, avoid double negative constructions (non-undesirable).
- Lines 244 till 247. The information in this paragraph is rather briefly presented, whereas this would be highly relevant to the topic of the review. It needs to be described more in detail.
- Lines 248 and further. There is inconsistency in the writing style: The authors refer to their study in bioRxiv, which is fine, however, first writing they (line 252) followed by we (line 254) is not consistent.
- Line 270. ‘many therapeutic hematopoietic malignancies’. What is meant by this? I think that the word therapeutic is misplaced here. It should be something like malignancies that are clinically treated or under therapy.
- I do miss a paragraph on various Cas proteins in use or under investigation (12, 13 for instance). See for example the review by Zhang in Biomedicine and Pharmacotherapy, 2021. It would add to the review, since there have been reports that CAS12 is more specific than CAS9. It would also add to the information on off-target effects. The advantage of the single guide RNA in the technology is missing as well, which is strange, because the crRNA and tracrRNA are mentioned. It can easily be added in the meant paragraph (lines 175 and further) or under future directions.
Author Response
Dear Biology editor and dear reviewer,
First of all, we would like to thank you for your kind consideration of our work as well as your efforts in organizing the review process for our manuscript. We also thank the reviewer for the critical suggestions in guiding us to further improve our manuscript. We are now submitting the revised manuscript (biology-1089994 – original) entitled “Future approaches for treating Chronic Myeloid Leukemia: CRISPR therapy” that we would like to be considered for publication as a review article. Hopefully, after reading our point-to-point response to review suggestion, you will concur with us that we made significant progress and have satisfactorily addressed all comments raised by the reviewer.
We have studied comments carefully and have made corrections which we hope to meet with approval. All amendments are highlighted in yellow in the revised manuscript which has the “Track Changes" function active so that changes are easily visible to the editor and reviewer.
To avoid movements of figures and number lines in the word document we have also submitted a PDF “named preprint” with all amendments highlighted in yellow.
Point by point response to the comments are as following:
Reviewer
The manuscript is well written.
The manuscript adds to the work described by this group previously and it intends to give an overview of the use of Crispr therapy in the CML field.
The balance is strongly towards CML description (approx. 6 pages) and less strongly on Crispr Cas technology (approx. last 2.5 pages).
It would certainly contribute to the field when the information on Crispr Cas application in CML would be more extensive and the emphasis would be a bit less on the BCR ABL biochemistry and physiology.
Author´s response
Thanks for this professional comments and valuable suggestions, we strongly agree with all your comments. We have extended the information about CRISPR Cas system and its application in human therapy and CML. Besides, following your suggestions we have added a new heading titled “New CRISPR-Cas systems and approaches” where different Cas proteins are described. New references of critical and great CRISPR papers have also been added. Figures 4 and 5 have been modified.
Reviewer
Details concerning the main text and figures.
In the abstract: change the last sentence to emphasize the focus on Crispr Cas technology and mention the reason why the technology is promising already here.
Author´s response
Thank you for your kind suggestion. We have added the following paragraph “The emergence of CRISPR technology can offer a definitive treatment based on its capacity to induce a specific DNA doble strand break. Besides, it has the advantage of providing complete and permanent oncogene knockout, while TKIs only ensure that BCR-ABL1 is inactivated during treatment. CRISPR/Cas9 cuts DNA in a sequence-specific manner making it possible to turn oncogenes off in a way that was not previously feasible in humans.”
Reviewer
Maybe addition of a few references, for instance Sternberg / Doudna on the off target effects of CAS9 would have added value for the complete story.
Author´s response
Thanks again for that suggestion. We have added a paragraph about off target effects….. “ The issue about CRISPR off-targets also needs to be resolved. CRISPR-Cas9 system induces DSBs at target sites in genomic DNA, but can also generate undesirable cleavages outside of on-target sites. Cleavage at off-target sites can trigger mutations which may result in the disruption of normal genes. Efforts to discover new Cas variants with high fidelity and a protospacer adjacent motif less restrictive than NGG will offer soon a solution” ……and we have added 3 references (109-111) included Sternberg et al. (lines 335-340)
We have also mentioned the off targets effects of DNA base editors (line 213-216) and its reference number 84.
Reviewer
Line 82, change enabling to ‘thereby presenting a target for clinical therapy’.
Figure 3: text is missing in the manuscript on the upper side of the figure; the figure may not be well placed or the text is really absent.
Line 127, the meaning of ‘destroyed’ is not clear in the context of the sentence (it is uncorrected or destroyed). Maybe remove the word destroyed.
Author´s response
Thanks a lot for the comments. Changes in line 82 have been done (now line 85). Figure 3 have been modified according your suggestion. The word destroyed have been replaced by unedited (now line 132).
Reviewer
Figure 4. The information is clear. However, when the legend explains the repopulation potential of modified LSC’s, they are eradicated and thus absent in the last picture, the visual information is a bit misleading, since these LSC’s are now depicted as normal cells. I get the point, but maybe the figure can be more in line with the text in the legend.
Figure 5. To my opinion, the green HDR panel is not so clear for the reader, maybe because of the arrows drawn on both strands. In addition, the figure has a few typos.
Author´s response
We completely agree with your comments and suggestions. Both figures have been modified and typos have been edited.
Reviewer
Line 168 and others; the word finally is used quite often, whereas it is not always the end of the information presented. For instance, the Crispr Cas biotechnological revolution only starts after the mentioned hallmarks. I think in this respect that the paper by Wright, Nunez and Doudna in Cell should be referred to and there are other great papers that are left aside. This is the main point of my comments: the Crispr Cas part can be stronger and more informative.
Author´s response
Thanks for this professional comments. In this sense we have added several refences and paragraphs, included the paper of Doudna in Cell (ref. 75). Crispr Cas part have been completely re-written according your comment, all these paragraphs with references have been added:
“Most of archaea and almost 50% of bacteria have an adaptive immune system to defend them against phage infection. This system is defined by a genomic locus with a series of short palindromic repeats separated by unique “spacers”, preceded by an AT-rich “leader” sequence and forming a cluster56. Francisco Mojica was the first researcher in 1993 to describe this matrix of tandem-repeated sequences working on the archaea Haloferax mediterranei57. Previously, a similar structure was described in Escherichia coli and he also spotted a connection with eubacteria58. Mojica coined the acronym of CRISPR (Clustered Regularly Interspaced Short Palindromic Repeats) in accordance with Ruud Jansen, who first used the term in print in 200259”
“They also studied the role of the CRISPR associated proteins (Cas) Cas7 and Cas9, suggesting that Cas7 was involved in generating new spacers and repeats and Cas9 in breaking the DNA64.”
“These crRNAs were responsible for CRISPR-based resistance and they can be transferred from a resistant to a naïve strain inducing resistance in the second. crRNAs drive the Cas9 protein to cleave the invader genome, and this find opened the door to direct the destruction of a DNA sequence like a restriction enzyme but in a specific addressable manner65–68. Importantly, the only requirement for Cas9 nuclease activity was the existence of a small PAM motif (protospacer adjacent motif) at the 3' end of the target sequence69–71. A single precise blunt-end cleavage event 3 nucleotides upstream of the PAM sequence was the consequence of the Cas9 nuclease activity.”
“To further simplify the system, Charpentier and Doudna showed that the two RNAs could function in vitro when fused into a single-guide RNA (sgRNA). Like Siksnys74, they showed that Cas9 could cut purified DNA in vitro and it could be programmed with custom-designed sgRNAs73”
“However, the question that remained to be addressed was whether the CRISPR system would also work in mammalian cells. By mid-2012, Feng Zhang designed a robust three-component system consisting of Cas9 (orthologs from S. thermophilus or S. pyogenes), tracrRNA, and a CRISPR array for targeting 16 sites in the human and mouse genomes. Zhang answered the question and he showed that it was possible to mutate one or several genes at the same time inducing indels by NHEJ. Besides, the CRSPR system could insert new sequences at the target site via HR when a repair template was added76,77”
“In 2006, the emergence of 'base editing' era, a new approach that enables the direct, irreversible conversion of one target DNA base into another in a programmable manner, without requiring DSBs or a donor templates was developed from David Liu lab. An engineered fusion of CRISPR/Cas9 with the cytidine deaminase enzyme has the ability to mediate the direct conversion of cytidine to uridine, thereby effecting a C→T (or G→A) substitution83. The door to correct a variety of point mutations relevant to human disease began to open. However, it was reported that these DNA base editors can cause substantial off-target editing in both genomic DNA and RNA, further studies will be performed to improve the selectivity of DNA base editors84”
Reviewer
Line 185. This is an opportunity to refer to the review of Wu in theranostics on Crispr systems in human genetic disease.
Author´s response
Reference added (ref 85). Thanks for suggestion.
Reviewer
Line 207. The relevance of Table 1 is not explicitly mentioned, it is only referred to in the text. I think it would be important to explain the information presented in the table.
Author´s response
Thanks again. We have added the following text “Several studies have recently performed to study the ability to disrupt the BCR/ABL1 oncogene, showing the CRISPR-Cas9 system as a therapy tool ready to reach clinical trials in a near future (Table 1).”
Reviewer
Line 242. Please, avoid double negative constructions (non-undesirable).
Author´s response
The sentence has been modified “…the disruption of the ABL1 non-rearranged allele did not trigger important consequences”.
Reviewer
Lines 244 till 247. The information in this paragraph is rather briefly presented, whereas this would be highly relevant to the topic of the review. It needs to be described more in detail.
Author´s response
The information has been completed with the following Text: ” Combinations of two sgRNAs targeting BCR intron 8 and ABL intron 1 regions were designed to induce a 133.9 kb deletion on BCR/ABL1 oncogene. This new strategy induced a frameshift alteration of the entire ABL1 DNA-binding domain. For any given sgRNA combination, electroporated K562 cells showed a significant decrease (∼85%) in clonogenic capacity in vitro and an increase in apoptosis. Importantly, they used cord blood-derived human hematopoietic progenitor hCD34+ cells the study whether a side effect was produced because of the CRISPR-Cas9 activity. In vitro analysis of targeted CD34 + cells revealed no difference in proliferation in long-term culture, suggesting no production of collateral cancer-driven genomic alterations. Besides, K562 transduced cell with an adenoviral vector carrying all the CRISPR system were subcutaneously injected into immunodeficient mice resulting in an 88% decrease in tumor size compared with control tumors.”
Reviewer
Lines 248 and further. There is inconsistency in the writing style: The authors refer to their study in bioRxiv, which is fine, however, first writing they (line 252) followed by we (line 254) is not consistent.
Author´s response
Thanks for noticed it. We have corrected this.
Reviewer
Line 270. ‘many therapeutic hematopoietic malignancies’. What is meant by this? I think that the word therapeutic is misplaced here. It should be something like malignancies that are clinically treated or under therapy.
Author´s response
Thanks again, reviewer are right. The sentence has been replaced by “…in hematopoietic malignancies that are clinically treated”.
Reviewer
I do miss a paragraph on various Cas proteins in use or under investigation (12, 13 for instance). See for example the review by Zhang in Biomedicine and Pharmacotherapy, 2021. It would add to the review since there have been reports that CAS12 is more specific than CAS9. It would also add to the information on off-target effects. The advantage of the single guide RNA in the technology is missing as well, which is strange, because the crRNA and tracrRNA are mentioned. It can easily be added in the meant paragraph (lines 175 and further) or under future directions.
Author´s response
Thanks for your kind suggestion. We agree with reviewer. To improve the quality of the review we have included this new section with references:
“New CRISPR-Cas systems and approaches
But Zhang's scientific contributions did not stop there and continued (and continue today) to revolutionize the field of the gene editing. In 2015, Zhang team reported a novel and more accurate CRISPR system from Acidaminococcus and Lachnospiraceae, with efficient genome-editing activity in human cells, the Cas12a (Cpf1)87.Cpf1 coding sequence is smaller than Cas9, it requires only one RNA (tracrRNA-independent ) and generates sticky-end DSBs that are less prone to non-homologous end joining (NHEJ) being ideal for precise gene editing. Given these advantages, Cpf1 system is more feasible to in vivo deliver and it could become a better gene editing tool than Cas988,89. A peculiar property of Cas12a was that it completely degraded single strand DNA molecules after its activation. By combining Cas12a ssDNase activation with isothermal amplification, Doudna team created a method named DETECTR (DNA endonuclease-targeted CRISPR trans reporter), which achieves attomolar sensitivity for DNA detection, showing its ability to detect human papilloma virus in patient samples in a rapid and specific manner90. In 2017, Zhang and col. again astonished the scientific community reporting a novel CRISPR system to target and edit RNA, the class 2 type VI CRISPR-Cas effector Cas13a91,92. The nuclease activity of this protein allows gene knockdown without genomic alteration. However, Cas13a cleaved all the RNAs around after becomes enzymatically active following the first single strand RNA break. Far from being a disadvantage, this promiscuous feature was used for DNA or RNA detection in diagnostics93. Specific high-sensitivity enzymatic reporter unlocking (SHERLOCK) was the first platform based on CRISPR-Cas13 systems for rapid and specific detection of viruses or mutations in patient liquid biopsy samples94. “
Information about off targets has been added as we mentioned above.
The information about the advantage of a single guide has been added in lines 193-196 as follow: “To further simplify the system, Charpentier and Doudna showed that the two RNAs could function in vitro when fused into a single-guide RNA (sgRNA). Like Siksnys74, they showed that Cas9 could cut purified DNA in vitro and it could be programmed with custom-designed sgRNAs73.”
We have tried our best revise our manuscript according to the comments. We would like to express our great appreciation to editor and reviewer for comments on our paper.
Looking forward to hearing from you.
Best regards
Manuel Sanchez-Martin
Corresponding author: adolsan@usal.es
WEBs:https://www.researchgate.net/profile/Manuel_Sanchez-Martin2
https://orcid.org/0000-0001-8370-1336

Reviewer 2 Report
This review manuscript from Vuelta et al showed CRISPR therapy strategy to CML. They summarized CML, translocations, and CRISPR systems and their usage. Generally, I enjoyed this review and learned a lot. But the manuscript was obviously overlooked by authors. I am afraid the all authors did not check the final manuscript. One figure was missing. Table is located at the wrong place. Contact information of authors are missing. I recommend authors to read instructions of authors for submission.
It is unclear to understand color coding of figure 1. Did author try to color codes with cell types or cancerous cells? What is the blue/orange/even green cells in the lymphoid cells? Please sort them and clarify in the legends.
Please change fluorescent in situ hybridization to fluorescence in situ hybridization
According to google, reverse transcriptase PCR hits 8 million results and reverse transcription PCR hits 80,000,000 results. I recommend authors should change transcriptase to transcription.
Figure 2A. I am not sure why authors illustrated chromosome 22 with dicentric or satellite-ish shape for the p-arm end. If not necessary please illustrate as chromosome 9.
Figure 4, It is confusing whether CML tumor cells (red spiky cells) is coming from orange or blue cells. Please clarify this in the legends. I am afraid red color will mislead readers.
Figure 5, fix DBS to DSB.
Where is figure6?? only legend in the manuscript....Same for table 1. But I found table 1 in page 9 in 2 pages later....
There is a strange empty space between line 238 and 240.
Author Response
First of all, we would like to thank the reviewer for the critical suggestions in guiding us to further improve our manuscript. We are now submitting the revised manuscript (biology-1089994 – original) entitled “Future approaches for treating Chronic Myeloid Leukemia: CRISPR therapy” that we would like to be considered for publication as a review article. Hopefully, after reading our point-to-point response to review suggestion, you will concur with us that we made significant progress and have satisfactorily addressed all comments raised by the reviewer.
We have studied comments carefully and have made corrections which we hope to meet with approval. All amendments are highlighted in yellow in the revised manuscript which has the “Track Changes" function active so that changes are easily visible to the editor and reviewer.
To avoid movements of figures and number lines in the word document we have also submitted a PDF “named preprint” with all amendments highlighted in yellow.
Point by point response to the comments are as following:
Reviewer #2
This review manuscript from Vuelta et al showed CRISPR therapy strategy to CML. They summarized CML, translocations, and CRISPR systems and their usage. Generally, I enjoyed this review and learned a lot. But the manuscript was obviously overlooked by authors. I am afraid the all authors did not check the final manuscript. One figure was missing. Table is located at the wrong place. Contact information of authors are missing. I recommend authors to read instructions of authors for submission.
Author´s response
Thanks for this professional comments and valuable suggestions. I am sorry in your doc the figure was missing and table in wrong position. We have got same problem. To avoid it, we have submitted a PDF document where you can track all changes made highlighted in yellow without that issue.
We agree with you in all your comments. Despite filiations of authors was present with the corresponding author email address, we have added all authors mail addresses.
Reviewer #2
It is unclear to understand color coding of figure 1. Did author try to color codes with cell types or cancerous cells? What is the blue/orange/even green cells in the lymphoid cells? Please sort them and clarify in the legends.
Author´s response
Thank you for your kind suggestion. We have modified the figure1 with more information in the drawn (naming the groups of cells) and completing the legend as follow: “Figure 1. Chronic myeloid leukemia clinical phases. A. Normal hematopoiesis characterized by the existence of hematopoietic stem cells with a controlled self-renewal and multipotency ability, resulting in balanced hematopoiesis between myeloid and lymphoid lineages. B. In the chronic phase the myeloproliferative differentiation pathway acquires an advantage and a massive myeloid expansion is produced. C. Blast crisis is characterized by a maturation arrest in the myeloid or lymphoid lineage. Newly accumulated genetic and epigenetic aberrations appear in LSCs and blast cells go out from bone marrow to peripheral blood.”
Reviewer #2
Please change fluorescent in situ hybridization to fluorescence in situ hybridization.
According to google, reverse transcriptase PCR hits 8 million results and reverse transcription PCR hits 80,000,000 results. I recommend authors should change transcriptase to transcription.
Author´s response.
Thanks for the comments. Changes are already done.
Reviewer #2
Figure 2A. I am not sure why authors illustrated chromosome 22 with dicentric or satellite-ish shape for the p-arm end. If not necessary please illustrate as chromosome 9.
Figure 4, It is confusing whether CML tumor cells (red spiky cells) is coming from orange or blue cells. Please clarify this in the legends. I am afraid red color will mislead readers.
Figure 5, fix DBS to DSB.
Where is figure6?? only legend in the manuscript....Same for table 1. But I found table 1 in page 9 in 2 pages later....
There is a strange empty space between line 238 and 240.
Author´s response.
We completely agree with your comments. Thanks for your help to improve the quality of this review, it is really appreciated.
Indeed chromosome 22 is poorly drawn and it seems that it has two centromeres, figure 2 has been replaced with a normal chromosome 22. Figure 4 has been modified for better understanding. Typos in Figure 5 have also been edited. Regarding Fig 6, we cannot understand how it has disappeared from the text, hopefully in the pdf attached you can all figures and legends in the correct site. Empty spaces between lines have been eliminated.
We have tried our best revise our manuscript according to the comments. We would like to express our great appreciation to editor and reviewers for comments on our paper.
Looking forward to hearing from you.
Best regards
Manuel Sanchez-Martin
Corresponding author: adolsan@usal.es
WEBs:https://www.researchgate.net/profile/Manuel_Sanchez-Martin2
https://orcid.org/0000-0001-8370-1336

Round 2
Reviewer 1 Report
Thanks for the revised manuscript, it has improved substantially and now the contents give adequate emphasis to the Crispr Cas part as used in the field of study. I have no further comments.
Reviewer 2 Report
Authors revised all my points.